# A registration strategy to characterize DTI-observed changes in skeletal muscle architecture due to passive shortening

**Melissa T. Hooijmans**[1,2,3], **Carly A. Lockard**[2], **Xingyu Zhou**[1,2,4], **Crystal Coolbaugh**[1], **Roberto P. Guzman**[2], **Mariana E. Kersh**[5,6,7], **Bruce M. Damon**[1,2,4,5,6,7,8,9,*]

1 Vanderbilt University Institute of Imaging Science, Vanderbilt University Medical Center, Nashville, Tennessee, United States of America, 2 Carle Clinical Imaging Research Program, Stephens Family Clinical Research Institute, Carle Health, Urbana, Illinois, United States of America, 3 Department of Radiology and Nuclear Medicine, Amsterdam UMC, University of Amsterdam, Amsterdam, The Netherlands, 4 Department of Biomedical Engineering, Vanderbilt University, Nashville, Tennessee, United States of America, 5 Department of Mechanical Science and Engineering, University of Illinois at Urbana-Champaign, Urbana, Illinois, United States of America, 6 Department of Biomedical and Translational Sciences, Carle-Illinois College of Medicine University of Illinois at Urbana-Champaign, Urbana, Illinois, United States of America, 7 Beckman Institute, University of Illinois at Urbana-Champaign, Urbana, Illinois, United States of America, 8 Department of Radiology and Radiological Sciences, Vanderbilt University, Nashville, Tennessee, United States of America, 9 Department of Bioengineering, University of Illinois at Urbana-Champaign, Urbana, Illinois, United States of America

* b.damon@carle.com

## Abstract

Skeletal muscle architecture is a key determinant of muscle function. Architectural properties such as fascicle length, pennation angle, and curvature can be characterized using Diffusion Tensor Imaging (DTI), but acquiring these data during a contraction is not currently feasible. However, an image registration-based strategy may be able to convert muscle architectural properties observed at rest to their contracted state. As an initial step toward this long-term objective, the aim of this study was to determine if an image registration strategy could be used to convert the whole-muscle average architectural properties observed in the extended joint position to those of a flexed position, following passive rotation. DTI and high-resolution fat/water scans were acquired in the lower leg of seven healthy participants on a 3T MR system in +20° and –10° ankle positions. The diffusion and anatomical images from the two positions were used to propagate DTI fiber-tracts from seed points along a mesh representation of the aponeurosis of fiber insertion. The –10° and +20° anatomical images were registered and the displacement fields were used to transform the mesh and fiber-tracts from the +20° to the –10° position. Student's paired t-tests were used to compare the mean architectural parameters between the original and transformed fiber-tracts. The whole-muscle average fiber-tract length, pennation angle, curvature, and physiological cross-sectional areas estimates did not differ significantly. DTI fiber-tracts in plantarflexion can be transformed to dorsiflexion position without significantly affecting the average architectural characteristics of the fiber-tracts. In the future, a similar approach could be used to evaluate muscle architecture in a contracted state.

**Data availability statement:** The fat/water MRI and unprocessed DTI data are available to any qualified investigator at an academic institution or private research organization who has current human subjects research ethics training certification and can describe a scientific use for the data. Interested parties can use the following link (https://redcap.link/AR073831_Data_Request) to initiate a Data Use Agreement.

**Funding:** BD acknowledges National Institute of Health (NIH) grants (https://grants.nih.gov/); National Institute of Health R01 AR073831 and National Institute of Health S10 OD021771. The sponsor did not play a role in the study design, data collection and analysis, decision to publish or preparation of the manuscript.

**Competing interests:** The authors have declared that no competing interests exist.

## Introduction

The diversity of human motion, including movements that vary in velocity, degree of precision, force, and duration, requires a broad range of outputs from the musculoskeletal system. The mechanisms for these varied outputs include neural control strategies (the activation patterns of multiple muscles; the motor unit recruitment and rate coding strategies within a muscle) and the activated muscles' metabolic, structural, and mechanical properties [1,2]. Several of the structural determinants of a muscle's mechanical properties include its gross morphology; its internal structure (or architecture, as represented by the pennation angle, length, and curvature of its fibers); and the structural and mechanical properties of the collagen-rich structures (such as tendons and aponeuroses) with which it interacts [3]. Skeletal muscle architecture can be non-invasively characterized using Diffusion Tensor Imaging (DTI), an MRI technique that measures the reduced and anisotropic diffusion of water molecules [4,5]. By measuring the anisotropic movement of water molecules, it is possible to derive information about the micro-structure and architecture of healthy and diseased muscle [6–23].

To date, DTI has primarily been applied in non-contracting, healthy muscle tissue. From such measurements, the microstructural and architectural properties of resting muscle have been characterized, including following passive joint motions [16,24]. Furthermore, these microstructural and architectural properties have been used as inputs to musculoskeletal models to predict changes in muscle function in healthy, diseased, and injured muscle [25–30]. DTI's intrinsic sensitivity to the three-dimensional aspects of muscle architecture and its potential for whole-muscle coverage and integration with other forms of functionally relevant MRI contrast create the possibility for using DTI along with other forms of MRI contrast to explore and answer previously unrealizable questions in the study of muscle structure-function relationships.

To realize such possibilities, DTI data would ideally be acquired during a muscle contraction; but this is challenging due to long scan times required for muscle DTI, the potential for motion artifacts, and signal voids that may occur when diffusion-weighted images are acquired during contraction [31–33]. This challenge is amplified by the fact that whole muscle DTI is required to provide a complete understanding of the architectural changes during contraction, and a whole-muscle DTI acquisition typically entails multi-stack acquisitions (which increases the total acquisition time in direct proportion to the number of stacks). Hence, to be able to obtain information about architectural properties in a whole muscle, during a contraction, a different approach is required.

Planar ultrasonography has been used in dynamic settings to characterize changes in architectural properties with a high temporal resolution, but this technique lacks the coverage required to visualize the whole muscle [34–37] and lacks MRI's broad range of soft tissue contrast. Recent work showed that high resolution displacement fields derived from MR image registration were able to accurately quantify tissue displacement and strain in skeletal muscle during isometric contractions [38,39]. Based on these findings, we hypothesize that we can use displacement fields derived from registration of high-resolution anatomical images in different ankle positions to transform muscle fiber-tracts from a plantarflexed to a dorsiflexed ankle position, without significant differences in the whole muscle's mean architectural properties. If so, this would provide critical proof of the concept that a registration-based approach could be similarly used to transform the DTI-observed skeletal muscle architecture patterns due to contraction.

The overall aim of this study was to evaluate if an image registration strategy could be used to convert the whole-muscle average architectural properties from an extended joint position to those of a flexed position. It is expected that the accuracy of the displacement fields, and

thus the transformation of muscle fiber-tracts, would depend highly on the quality of the registration. Therefore, we first optimized the registration parameters of the anatomical images; thereafter used interpolation to transform fiber-tracts from a plantarflexed to a dorsiflexed ankle position; and finally determined whether the original tracts' mean architectural properties differed from those of the transformed tracts.

## Materials and methods

### Study participants

Seven healthy volunteers (5 men; Age: 25.1 ± 2.7 yrs. with range 23 – 31 yrs.) provided written informed consent to participate in this IRB-approved study (Vanderbilt University Medical Center IRB committee). The inclusion criteria were healthy men and woman in the age range of 18 - 40 years. Exclusion criteria included the inability to provide informed consent, MRI contraindications, claustrophobia, physician-diagnosed muscle disease or other disease affecting muscle function, an inability to exercise safely, drug and alcohol use to the point of intoxication more than twice a week, and smoking tobacco. All participants were asked to abstain from alcohol (24 hours prior to the examination), caffeine (6 hours prior to the examination), and moderate or intense exercise (24 hours prior to the examination).

### MR examination

MR datasets were acquired in the right lower leg on a 3 Tesla MR System (Philips Elition; Best, the Netherlands) using a 16-element receiver coil (anterior) and the 10-element receiver coil built into the patient table (posterior). The participants were positioned supine, feet-first in the MR scanner with the right leg as close as possible to the center of the bore (S1 Fig). The foot was placed in an MR compatible exercise device that allowed the foot to be positioned at ankle angles ranging from −15° degrees dorsiflexion to +25° degrees plantarflexion. The anterior coil was placed on top of the legs and supported with foam pillows and fixation bands to ensure that the coil covered the full lower leg, did not compress the muscle, and did not move during the change of ankle position from plantarflexion (+20°) to dorsiflexion (−10°). MRI data were first acquired with the ankle passively held at +20° and again after the ankle was passively rotated to −10°. These values were selected because they represent a functionally relevant range of motion, while maintaining comfort for the participants.

The MR examination consisted of:

I. Chemical shift-based water-fat separation scan for anatomical reference and muscle and aponeurosis segmentation. (3D FFE; mDixon-Quant; TR/TE/ $\Delta$ TE/FA, 210ms/1.01ms/0.96ms/3°; voxel size, $1 \times 1 \times 1.75 mm^3$; number of excitations ($N_{EX}$), 1; acquired matrix, $192 \times 192$; number of slices, 176; no slice gap; SENSE, 2; and scan duration, 36.2 seconds).

II. Two diffusion-weighted acquisition to assess muscle fascicle architecture (SE-EPI; TR/ TE, 4800ms/53ms; 24 directions; b-values, 0 (5), 450 (19) s/mm²; recon voxel size, $1 \times 1 \times 7 mm^3$; $N_{EX}$, 4; acquired/reconstructed matrices, $96 \times 96/192 \times 192$; number of slices, 24; no slice gap, SENSE, 1.7; combination of three fat suppression techniques: SPectral Adiabatic Inversion Recovery (SPAIR) and Slice Selected Gradient Reversal (SSGR) for the aliphatic lipid peaks and a SPIR pulse for the olefinic peak [40]; scan duration, 725 seconds). The DTI data were acquired in two transverse stacks with 28 mm overlap, fully covering 308 mm in the superior-inferior direction with a field of view (FOV) of $192 \times 192 mm^2$.

For each type of sequence, the superior end of the slice stack was positioned at the level of the tibial plateau and the image stacks were rotated to align their midpoint with the tibia bone.

## Data-analysis

All data-analysis was performed in MATLAB R2019 (The MathWorks, Natick MA) using the publicly available MuscleDTI_Toolbox [41] and additional custom written scripts. The full workflow of the study is shown in Fig 1.

### VOIs and aponeurosis mesh

For each ankle position, masks of the Tibialis Anterior (TA) boundaries and aponeurosis were manually drawn in ITK-snap (version 3.8.0; www.itksnap.org) [42] on the high resolution mDixon-Quant water image. The manually segmented aponeurosis masks were converted to a low-resolution seed-point mesh (mesh size: 30 × 20) using *define_roi()*. An additional high-resolution seed-point mesh (size 100 × 50) was also defined for visualizing the outcomes of the optimized method.

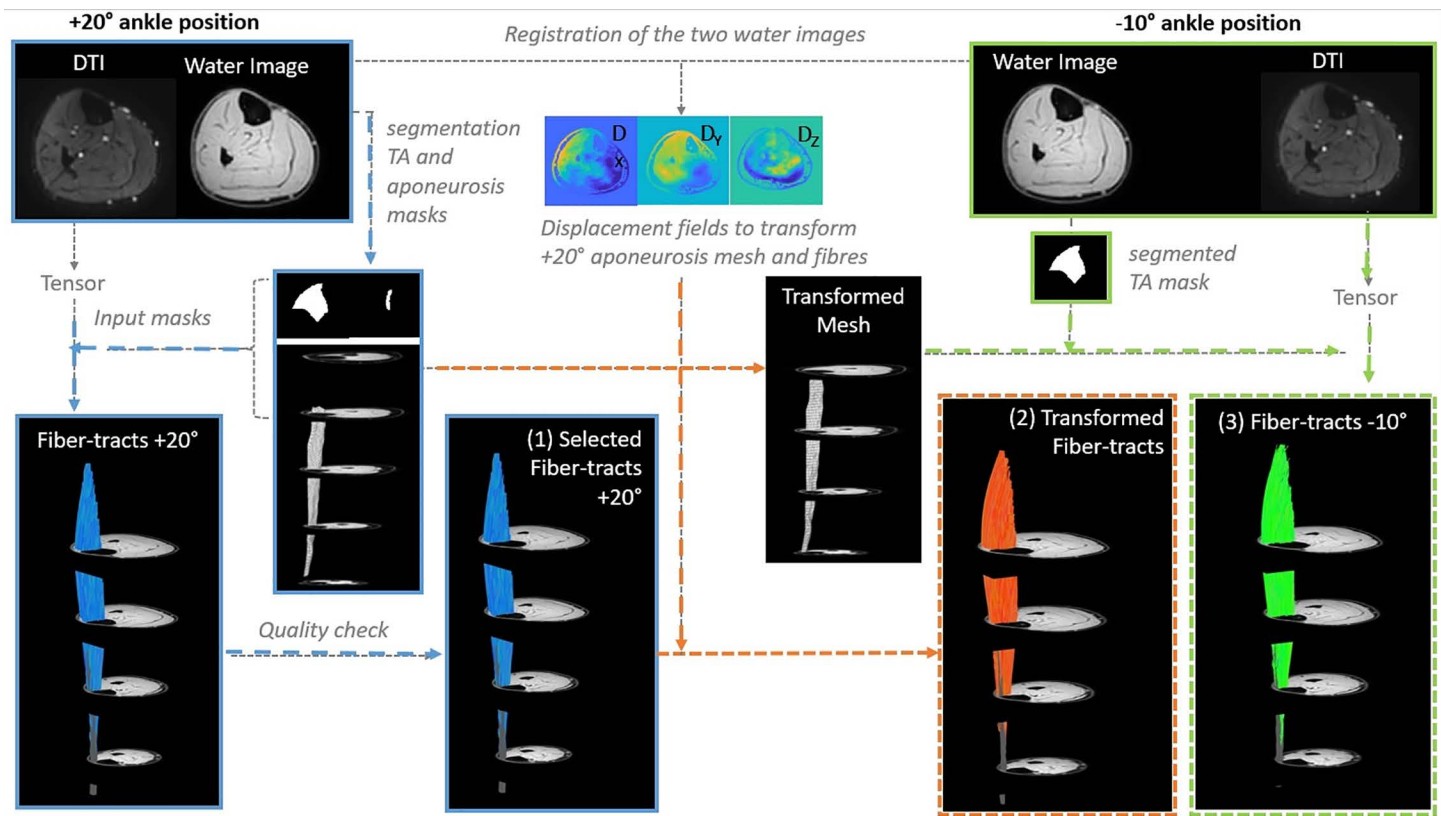

**Fig 1. Overview of the analysis workflow.** MR data acquired in two ankle positions, +20° (blue outline) and −10° (green outline). Anatomical images in the two ankle positions were manual segmented to derive masks for the TA aponeurosis (−10°) and TA (−10° and +20°) muscle. After the tensor calculation (+20°), fiber tracking was performed using the TA mask and aponeurosis mesh (+20°) as input. Resulting in the (1) fiber-tracts for the +20° position. Displacement fields derived from registration of the anatomical images (+20° and −10°) were used to transform fiber-tracts and the aponeurosis mesh from +20° to −10° ankle position (orange arrows), leading to the (2) transformed fiber-tracts. The transformed aponeurosis in combination with the manually segmented TA muscle mask (−10°) and tensor (−10°) (green arrows) are used to initiate tracking in the −10° position generating the (3) fiber-tracts for the +20° position. These inputs are required to allow direct comparison between (2) transformed (orange dotted outline) and (3) original fiber-tracts (green dotted outline).

## Optimization of anatomical image registration

The high-resolution anatomical scans from the +20° ankle position was registered to the anatomical scan from the −10° position using MATLAB's *imregdemons()* function (2000 iterations, 4 pyramid levels and AccumulatedFieldSmoothing of 1.0; Fig 1). This function uses a 2D multi-slice registration based on a Demons algorithm [43], resulting in displacement fields for the X, Y and Z directions. A variety of registration inputs were explored, including various contrasts (*2 settings; water image and out-of-phase images*); slice thickness (*3 settings; 7 mm, 3.5mm and 1.5mm*); normalization (*2 settings; normalized images and non-normalized images*); and masking options (*2 settings; full mask lower leg; mask for the TA muscle*). Image normalization was performed by dividing the signal intensities in the original water or out-of-phase image by the maximal signal intensity in the image slice. Masking of the full lower leg was based on an empirically determined signal intensity threshold. For the TA-only mask, the manually segmented boundary mask was used.

The registration displacement fields were smoothed using a Savitzky-Golay filter with a polynomial order of 2 and frame length of 2, as implemented in the MATLAB function *sgolay()*. This registration field was used to transform the muscle boundary mask and the seed-point mesh from +20° ankle position to the −10° position by 3D interpolation (MATLAB's function *interp3*) to the −10° ankle position; see Fig 1. For each set of registration inputs, the outcomes were assessed by calculating the Sorenson Dice Similarity Coefficient (DSC), a similarity quotient that ranges between zero and one and is calculated as:

$$DSC = A \cap B / (|A| + |B|)$$

where A is set of points in the mask of image A, B is set B, ∩ indicates the intersection of the sets, and | | indicates the size of the set. The DSC was calculated for the muscle mask (DSC$_{mask}$) and aponeurosis (DSC$_{apo}$) mask. In addition, the registration outcomes were assessed using the Hausdorff distance (D$_H$) and Euclidean distance (D$_E$) for the aponeurosis mesh. The optimal registration approach was defined as the method with the best ordinal ranking across all outcome assessments.

## Pre-processing of diffusion images

The diffusion images were registered to the scanner-reconstructed water images of the Dixon series using the *imregdemons()* function. Then the diffusion data were denoised using an anisotropic smoothing algorithm [44,45] using a noise level of 5% [46]. The diffusion tensor was estimated in each voxel from the denoised diffusion data using weighted least squares fitting, and the tensor's eigenvalues and eigenvectors were found using singular value decomposition, as described previously [41].

## Fiber-tracking

The diffusion tensor field, seed-point mesh, and muscle boundary mask were used as inputs for fiber-tracking. Fiber-tracts were initiated from points along the seed-point mesh and propagated using 4th-order Runge-Kutta integration of the principal eigenvector at a step size of 1.0 pixel-width. Stopping criteria included either 1) reaching the muscle mask boundary or 2) if two consecutive data points had a Fractional Anisotropy value < 0.1 or > 0.4 or a trajectory angle between consecutive tracking steps of > 30°. Three sets of fiber-tracts were generated:

1. Fiber-tracts were generated for the +20° ankle position after initiation from the seed-point mesh originally defined for this position (the original fiber-tracts from the +20° ankle position).

2. Fiber-tracts were generated for the −10° ankle position by transforming the fiber-tracts from the +20° position to their predicted locations at an ankle angle of −10° (the transformed fiber-tracts from −10°). The transformation was performed as described above.

3. Fiber-tracts were generated for the −10° ankle position after initiation from the seed-point mesh that was transformed from the +20° position (the original fiber-tracts from the −10° position). The transformed mesh was used to enable direct comparison between the transformed and original datasets.

## Architectural quantification and goodness filtering

Prior to architectural quantification of the fiber-tracts, the row, column and slice positions were smoothed using 3$^{rd}$ order polynomial fitting [47,48]. Architectural tract properties, including pennation angle (θ; [7]), fiber-tract length (L$_{FT}$), and curvature (κ; [47]), were quantified for each tract in all subjects using the *fiber_quantifier()* function. Pennation angle was defined as the complement to the angle formed by the normal vector to the aponeurosis at the seed point and the position vectors. The fiber-tract length is calculated by summing the inter-point distances, and the curvature is calculated using a discrete implementation of the Frenet-Serret equations. The *fiber_goodness()* function was used to exclude fiber-tracts if their Z-positions did not increase monotonically; if their mean curvature was > 40 m$^{-1}$, their mean pennation angle was > 40°, and/or their length was < 10 mm or > 98mm; or if there were local outliers in any of these architectural properties. The maximum length criterion was set at the average fascicle length plus twice the standard deviation reported in an ultrasonography study [54]. For pennation angle and curvature, values were selected to ensure the exclusion of physiologically unrealistic values or otherwise implausible data. In addition, the muscle volume (V$_{M}$), mean (L$_{FT}$), and θ were used to calculate the Physiological Cross-sectional Area (PCSA) as [49]: PCSA = V$_{M}$ * cos (θ)/ L$_{FT}$, using L$_{FT}$ as a proxy for fascicle length.

## Fiber-tract similarity measures

To compare the original and transformed −10° fiber-tract outcomes, we determined the differences in the mean pennation angle (Δθ), fiber-tract length (ΔL$_{FT}$) and curvature (Δ$\underline{k}$) for the deep compartment, superficial compartment, and whole muscle, with the superficial and deep compartments delineated by the internal aponeurosis. Additionally, the similarity (S$_{i}$) was calculated between the transformed and its corresponding original fiber (similar mesh starting point) using: S$_{i}$ = R$_{CS}$ · e$^{(-D_E/C)}$ [8,50], where R$_{CS}$ is the corresponding segment ratio, D$_{E}$ is the mean Euclidean distance between corresponding points, and C is a weighting factor (here set to 1.50 mm). We used the median S$_{i}$ to characterize this property on a whole-muscle level. As exploratory analyses, we also compared the differences in architectural properties and S$_{i}$ at the individual fiber-tract level.

## Statistical analysis

All statistical analysis were performed using SPSS (IBM SPSS Statistics 28.01.1.1, Armonk, NY: IBM Corp). Data were checked for normality using Shapiro – Wilk test. Paired *t*-tests or the non-parametric version of the *t*-test were used to evaluate the difference in mean architectural properties (θ, L$_{FT}$, κ, and PCSA) between the original and transformed fiber-tracts in the deep compartment, superficial compartment, and whole muscle. Pearson correlation and Bland-Altman analysis were used to exploratory evaluate architectural properties on the fiber-tract level. Furthermore, Pearson correlations were used to explore the relation between the registration outcome measures (DSC$_{mask}$, DSC$_{apo}$, D$_{H}$ and D$_{E}$) and the similarity measures (Δθ, ΔL$_{FT}$, Δ$\underline{k}$ and S$_{i}$).

## Results

### Data quality

All MR datasets were complete and without observable fat and/or motion artifacts.

### Registration approaches

The DSC, $D_H$ and $D_E$ for the muscle mask, aponeurosis mask and aponeurosis mesh using the different registration approaches are shown in Table 1. Overall, the non-normalized out-of-phase (TE = 1.01ms) images with a slice thickness of 3.5mm resulted in the lowest ordinal scale and were selected to transform the aponeurosis mesh and fiber-tracts. One of the datasets did not register correctly, as assessed by visual inspection by MH and low DSC values for the muscle and the aponeurosis masks. Thus, for the comparison of architectural properties and correlation analysis we excluded this dataset.

### Architectural properties

Representative original and transformed datasets, using the high-resolution seed-point mesh definition and the optimized registration criteria from Table 1, are shown in Fig 2. Data were normally distributed and analyzed using a paired t-test. No significant differences were detected between the original and transformed fiber-tracts for the mean values of θ, $L_{FT}$, and $\kappa$ in the deep compartment (p-value θ: 0.64, $L_{FT}$: 0.74, and $\kappa$: 0.39), superficial compartment (p-value θ: 0.01, $L_{FT}$: 0.44, and $\kappa$: 0.92), and whole TA muscle (p-value θ: 0.13, $L_{FT}$: 0.72, and $\kappa$: 0.52)) (Table 2 and Fig 3). No significant differences were detected in PCSA between the original and transformed fiber-tracts in the whole TA muscle (p-value: 0.454). On an individual fiber-tract level variations were observed in the agreement in pennation angle and fiber-tract length between the original and transformed fiber's for both a high and a low similarity dataset (S2 Fig). A significantly higher pennation angle ($p = 0.004$) was found in −10° ankle position compared to + 20°, while no differences were detected in fiber-tract length and curvature between the two ankle positions ($p > 0.15$) (Fig 4).

### Registration quality measures in relation to fiber similarity measures

On a whole muscle basis, no significant correlations were observed between $S_i$ and $\Delta\theta$, $\Delta L_{FT}$, or $\Delta\kappa$ and the registration quality measures ($D_H$ (r = −0.41; p = 0.42), $D_E$ (r = 0.08; p = 0.91), $DSC_{apo}$ (r = 0.64; p = 0.18) and $DSC_{mask}$ (r = −0.65; p = 0.18)) and $\Delta\theta$ (r = −0.42; p = 0.42), $\Delta L_{FT}$ (r = −0.26; p = 0.66) (Fig 5) or $\Delta\kappa$ (r = −0.46; p = 0.76). Similarly, no clear pattern was displayed between $\Delta\theta$ and $\Delta L_{FT}$ and the registration quality measures (Pearson R absolute range = 0.30–0.67; $p > 0.102$) (S3 and S4 Figs). Within subjects, the similarity values ranged widely (between

**Table 1. Evaluation of the registration approaches.**

| Output | Water Image (7 mm) | OP Image (7mm) | OP Image (3.5mm) | OP Image (1.75mm) | Image Mask | TA mask |
|---|---|---|---|---|---|---|
| $DSC_{mask}$ | 0.91 ± 0.005 | 0.91 ± 0.001 | **0.92 ± 0.002** | 0.91 ± 0.004 | 0.90 ± 0.002 | **0.92 ± 0.002** |
| $DSC_{apo}$ | 0.76 ± 0.05 | 0.78 ± 0.05 | 0.74 ± 0.05 | 0.76 ± 0.05 | 0.77 ± 0.05 | **0.78 ± 0.02** |
| $D_H$ mesh | 41.4 ± 8.5 | 20.3 ± 6.9 | **17.2 ± 4.1** | 17.5 ± 3.5 | 40.8 ± 7.3 | 40.4 ± 7.1 |
| $D_E$ mesh | 3.6 ± 1.4 | 2.8 ± 1.1 | **2.3 ± 0.9** | 2.3 ± 0.9 | 3.5 ± 1.3 | 2.9 ± 0.8 |
| Ordinal output | 5 | 4 | **1** | 3 | 6 | 2 |

Data table showing the average and standard deviation of the DSC, Hausdorff distance ($D_H$), Euclidean distance ($D_E$) and overall ordinal grading for the muscle mask, aponeurosis mask and aponeurosis mesh using a selection of the different registration strategies. In bold text is shown the best approach per category. Note the best strategy from among those tested: Non-normalized Out-of-phase images with a slice thickness of 3.5mm.

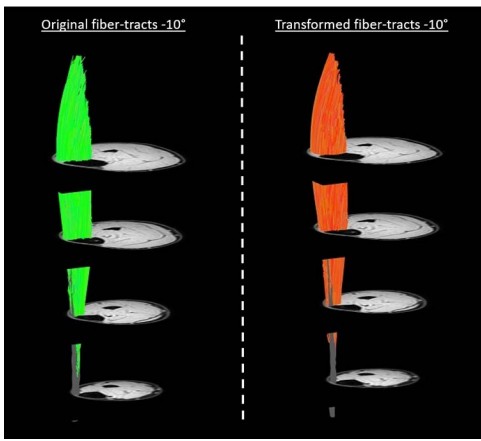

**Fig 2. Original and transformed fiber-tracts.** The original fiber-tracts (green) and aponeurosis mesh are shown on the left side and the corresponding transformed fiber-tracts (orange) and aponeurosis mesh on the right side for a representative dataset. For visualization purposes we used a high-density mesh (size $100 \times 50$).

**Table 2. Architectural properties of the original fiber tracts in the −10° and +20° ankle position and the transformed fiber tracts.**

| Outcome measure | Ankle position | Full TA muscle | Deep compartment | Superficial compartment |
|---|---|---|---|---|
| **Angle (°)** | **Original +20°** | $6.43 \pm 1.0°$ | $6.4 \pm 1.5°$ | $6.5 \pm 1.2°$ |
| | **Original −10°** | $9.1 \pm 1.4°$ | $10.7 \pm 3.3°$ | $10.0 \pm 3.5°$ |
| | **Transformed −10°** | $7.3 \pm 1.7°$ | $5.9 \pm 1.2°$ | $7.8 \pm 1.4°$ |
| **Length (mm)** | **Original +20°** | $51.5 + 4.51$ | $56.4 \pm 13$ | $59.2 \pm 8.8$ |
| | **Original −10°** | $49.9 \pm 4.25$ | $52.3 \pm 8.6$ | $47.7 \pm 7.6$ |
| | **Transformed −10°** | $48.3 \pm 10.8$ | $50.7 \pm 13.7$ | $43.9 \pm 9.8$ |
| **Curvature (m⁻¹)** | **Original +20°** | $7.2 \pm 0.7$ | $7.2 \pm 1.6$ | $6.9 \pm 0.2$ |
| | **Original −10°** | $8.7 \pm 1.4$ | $7.5 \pm 3.0$ | $9.8 \pm 2.1$ |
| | **Transformed −10°** | $9.8 \pm 3.2$ | $9.8 \pm 4.5$ | $9.8 \pm 2.1$ |

Data table showing the average and standard deviation pennation angle, fiber-tract length and curvature values for the original fiber-tracts in the dorsiflexed −10° ankle position, plantarflexed +20° ankle position and the transformed fiber-tracts. There were no significant differences between original and transformed fiber tracts.

0.03–0.77). Typical example fiber-tracts for a low and high similarity cases are shown in Fig 6. The average per-subject similarity value ranged between 0.14 and 0.30 across the sample. We did not find a clear pattern for the comparison of $S_i$ and $\Delta\theta$, $\Delta L_{FT}$ on a fiber-tract basis for both the highest and lowest averaged similarity score (S5 Fig).

## Discussion

We showed that DTI-determined muscle fiber-tracts can be transformed from a plantarflexed ankle position to the dorsiflexed ankle position using registration of high-resolution anatomical images. On the whole-muscle and whole-compartment scales, the original and transformed fiber-tracts did not differ in architectural characteristics, i.e., mean length, pennation angle, curvature, and PCSA. This finding emphasizes the potential of this approach for analyzing compartmental or whole-muscle variations in architectural properties during passive movements about a joint. However, there was variability in the quality of the outcomes across participants and across fiber-tracts within a participant. The applicability of this method therefore depends on the intended application of the data.

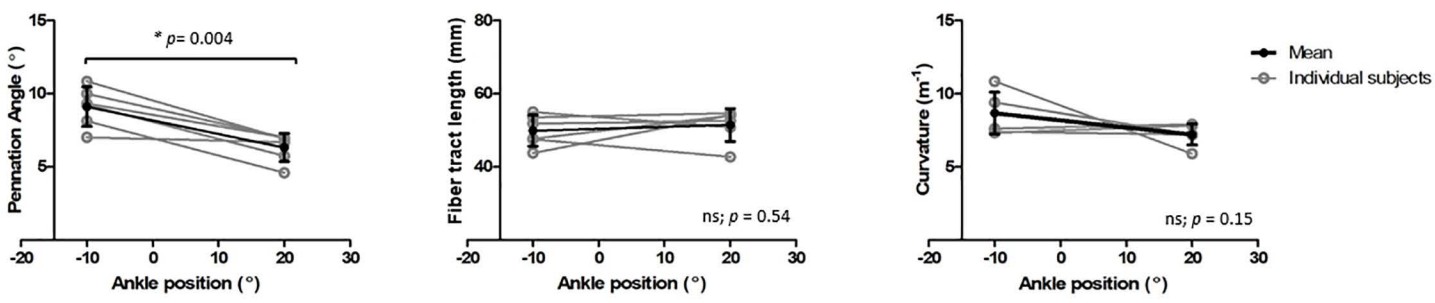

**Fig 3. Architectural properties of the original and transformed fiber-tracts.** Line plots showing the pennation angle (left), fiber-tract length (middle) and fiber-tract curvature for the original and transformed fiber-tracts in the full TA muscle (top row), the deep compartment of the TA muscle (middle row) and the superficial compartment of the TA muscle (bottom row). In black the mean and standard deviation for all the subjects and in light grey the individual subjects. Significant differences are indicated with an asterisk, and *p*-values are noted in the figure.

**Fig 4. Pennation angle and fiber tract lengths of the original fiber-tracts in the two ankle positions.** Line plots showing the pennation angle (left) fiber-tract length (middle) and curvature for the original fiber-tracts in −10° and +20° ankle position in the full TA muscle. In black symbols are shown the mean and standard deviation for all the subjects and in light grey symbols are shown the individual subjects' data. Significant differences are indicated with an asterisk and *p*-values are noted in the figure.

To our knowledge, this is the first study exploring the use of high-resolution displacement fields to transform DTI-determined muscle fiber-tracts between joint angles. The original and the corresponding transformed fiber-tracts showed similar mean architectural properties at the whole-muscle and compartmental scales. These properties were also in the same range

## Registration quality measures

## Similarity measures

Plots showing:
- Dice aponeurosis vs $S_i$ — p:0.18; r:0.64
- Dice mask vs $S_i$ — p:0.18; r:-0.65
- $\Delta PA$ (°) vs $S_i$ — p:0.42; r:-0.42
- $D_H$ vs $S_i$ — p: 0.42; r: -0.41
- $D_E$ vs $S_i$ — p:0.91; r: 0.08
- $\Delta L_{ft}$ (mm) vs $S_i$ — p:0.66; r:-0.26

**Fig 5. Correlation analysis between the Similarity index, similarity measures and registration measures.** Dot plots showing the correlation between similarity index ($S_i$), the tracts similarity measures and registration quality measures. The black dots represent the individual participants. $D_H$: Hausdorf Distance, $D_E$: Euclidean Distance; $\Delta\theta$: the difference in pennation angle between the original and transformed fiber-tracts, $\Delta L_{FT}$: the difference in length between the original and transformed fiber-tracts.

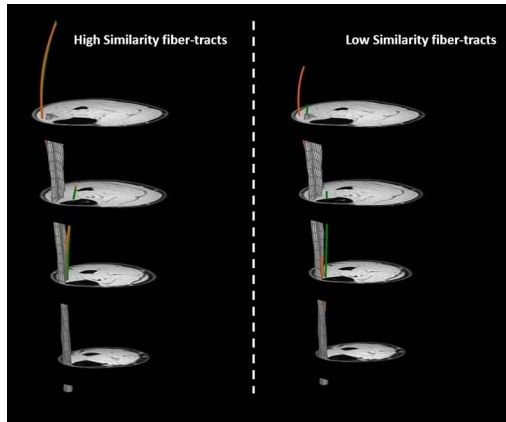

**Fig 6. Visualization of the transformation of high and low similarity fiber-tracts.** Visualization of variations in the quality of transforming muscle architecture by highlighting two individual fiber-tracts (one with a more proximal aponeurosis attachment and one with a more distal aponeurosis attachment). In green the original fiber-tracts and in orange the corresponding transformed fiber-tracts for one representative subject. The image on the left shows original and transformed fiber-tracts with high similarity ($S_i$ = 0.53 & 0.55) and the image on the right displays original and transformed fiber-tracts with low similarity ($S_i$ = 0.03 & 0.07).

as pennation angles and fascicle lengths previously reported in the TA muscle using DTI [7,15,16,51–53], ultrasonography [54–57], and cadaver [58,59] studies. When comparing the original and transformed tracts from the −10° position, we did observe compartment-specific variations in the fiber-tract lengths, but these did not rise to the level of statistical significance.

Besides non-invasively reflecting anatomy, muscle fiber-tract architecture is also used to predict function [3,60] and to detect changes in architectural properties between conditions. Indeed, the PCSA [61] is considered one of the key predictors of muscle force production; therefore, not finding changes in PCSA between the transformed and original fiber-tracts is very relevant with respect to the use of this method in musculoskeletal modelling [25–30]. Additionally, we observed a smaller pennation angle in the full TA in the plantarflexed (muscle lengthening) (+20°) compared to dorsiflexed (muscle shortening) (−10°) ankle position, which is consistent with physiological predictions based on the relative compliance of resting muscle vs. tendinous structures and previous observations that showed a smaller pennation angle in plantarflexion (+15 and + 30°) compared to dorsiflexed (−15°) position [54,62,63]. Interestingly, we did not observe a clear pattern for fiber-tract length on a group-level basis, and both longer and shorter fiber-tracts were seen across participants. This was contrary to both physiologically-based predictions and prior work on the full TA muscle that showed a reduced fascicle length in dorsiflexed position (−15°) compared to plantarflexed (+30°) ankle position [24,64]. Possible causes for the shorter fiber-tracts are muscle tendon complex lengthening, or some inconsistency or error in fiber-tracking due to a small mismatch in registration between the DTI and anatomical data. Even though physiological predictions suggested higher curvature in the dorsiflexed position, we did not observe a clear difference in this property at the group level. The failure to observe a significant change could have occurred due to insufficient statistical power for the study or if the curvature estimation method was insufficiently precise. Additional work may be necessary to improve the accuracy and precision of this method.

The quality of the transformation of fiber-tracts highly depends on the performance of the registration, in this case using the Demons algorithm. Previously, the Demons algorithm has been used and validated to accurately quantify displacement and deformations in skeletal muscle tissue [38,65], pelvic floor, lung and cortical bone tissue [66,67]. In addition, the reliability of this algorithm has been tested by imposing known deformation to an image set [38] and indicated that the measured strains are conservative estimators of the local deformations. To establish good quality transformation of fiber-tracts, we evaluated a variety of registration inputs to optimize displacement fields derived from the Demons algorithm and showed that the out–of–phase (OP) images with an intermediate slice thickness led to the overall best registration result. This may be due to the higher internal contrast between anatomic structures in these images compared to the water-only images, the lower leg mask, and muscle mask. The difference in registration quality between the three slice thicknesses was minimal and may indicate that this slice thickness is sufficient to sample any foot-head variations in structure, at least in the case of this muscle in healthy adults. The in-plane resolution of the anatomical images used for registration was similar to the resolution of the DTI data. Future work could determine if higher in-plane resolution could benefit the registration further.

In addition to comparing the compartmental and whole-muscle architectural properties of the entire dataset, similarity and architectural measures were evaluated at the fiber-tract level. The Bland-Altman analysis clearly visualized variations in similarity in architectural properties on an individual fiber-tract level, independent of a high or low similarity analysis. This finding is consistent with the correlation analysis which displayed clearly that tracts with both high and low similarities could have very similar architectural properties and that tract pairs with low similarity values could exhibit either high or low levels of agreement among their architectural

properties (S5 Fig). This suggests that the similarity index as used here may not fully assess fiber-tract likeness. This may be because the similarity measure accounts for both the amount of length overlap of two tracts and the distance between their corresponding points. Depending on the value chosen for the weighting coefficient, differently shaped but adjacent fiber-tracts and overlapping but more distant fiber-tracts can have the same similarity score [8,50,68]. Also, our registration quality measures did not show convincing correlations with our fiber similarity measures. This was contrary to our expectations, since we optimized according to these measures; however, this evaluation may have been impacted by the low number of datasets, the imperfect nature of this similarity index as an outcome measure, and small dynamic range of some of the registration outcome parameters. The problem of low dynamic range is exacerbated by excluding the dataset that did not register properly. In the next development phase of the registration-based approach to transforming fiber-tracts, it may be useful to expand the calculation of the similarity measure to also include differences in architecture between the tracts or include other structural features as terms in the similarity calculation.

The future goal of this approach is to explore muscle architecture during contractions. To address that goal, DTI data would be acquired in relaxed position and then transformed to its contracted state using anatomical images acquired at rest and during contraction. This approach necessarily rules out the possibility to compare fiber-tracts according to the similarity measure or the observed architectural properties, making validation less straightforward. From pilot data, we know that smaller displacements occur from relaxed to contracted state in a fixed ankle angle isometric contraction, suggesting that registration of these type of datasets should be less challenging. Furthermore, the displacement fields derived from the registration have also been used to quantify strain values in muscle; these strain values corresponded well with values measured with other techniques (MR-tagging and PC-MRI) [39,69]. The amount of correspondence between strain values derived from displacement fields and measured with other MR techniques during active contractions could possibly function as justification in active situations. All combined, this study emphasizes the potential of this approach for observing whole-muscle or compartment changes in architectural properties during isometric muscle contractions.

This study has several limitations. First, it is a proof-of-concept study, in a limited number of young, healthy participants which makes detecting meaningful differences challenging. Consequently, no difference between the original and transformed fiber-tracts does not automatically indicate the accuracy of this registration-based approach; rather, this work primarily showcases its potential. Further, the translatability to other populations (such as aging and neuromuscular diseases) should be explored further. However, some of the pathological features known to characterize aging and diseased muscle, i.e., the replacement of muscle tissue by fat, result in more contrast in the images, potentially benefitting the registration and the transformation of muscle fiber-tracts. Additionally, how well the fiber-tracts can be transformed depends highly on the performance of the registration. Here we only evaluated a narrow band of data for which the registration worked well and therefore may have overestimated the performance of the approach. Despite the relatively high DSC indices and low $D_H$ and $D_E$ we found using our registration approach, alternative or novel registration strategies or approaches could improve the outcomes from this study even further. We also note that the method did not perform as well on the fiber-tract level as it did on the compartmental and whole-muscle levels, which again suggests possible roles for alternative or novel registration strategies. Lastly, we used the fiber-tracts generated in −10° ankle position with the transformed mesh as ground truth. However, noise and other image quality issues will affect the performance of DTI tractography [64,70,71], perhaps leading to an underestimation of the performance of our approach.

In summary, we showed that muscle fiber-tract architecture from one ankle position can be transformed in the other ankle position using registration derived displacement fields. Whole-muscle architectural characteristics, i.e., fiber-tract length, pennation angle, curvature, and physiological cross-sectional area of the original and transformed fiber-tracts did not differ significantly on a group-level basis. Consequently, this approach to transform muscle architecture is very promising, and our next step will be to evaluate if muscle architecture can be reconstructed in a contracted state using a similar approach.

## Supporting information

**S1 Fig. Schematic overview of the positioning in the MRI scanner for the two ankle positions.** In green is shown the anterior receive coil positioned on top of the lower extremities. (TIF)

**S2 Fig. Bland-Altman Plots showing the similarity between the original and transformed fiber-tracts for pennation angle ($\Delta\theta$) and fiber-tract length ($\Delta L_{FT}$), on a fiber tract level, for the dataset with on average the highest similarity value (left) and for a dataset with on average the lowest similarity (right) value ($S_i$ = 0.14 & 0.31).** Note differences in Y-axis scales between the left and right panels. (TIF)

**S3 Fig. Dot plots displaying the correlation between the registration quality measures and the difference in pennation angle ($\Delta\theta$) between the original and transformed fiber-tracts for each of the participants (black dots).** (TIF)

**S4 Fig. Dot plots displaying the correlation between registration quality measures and the difference in fiber-tract length ($\Delta L_{FT}$) between the original and transformed fiber-tracts for each of the participants (black dots).** (TIF)

**S5 Fig. Scatterplots showing the relation between the similarity ($S_i$) and the difference in pennation angle ($\Delta\theta$) and fiber-tract length ($\Delta L_{FT}$) between the original and transformed fiber-tracts for each of the individual fiber-tracts for the dataset with the highest and lowest averaged similarity value.** (TIF)

## Acknowledgments

The authors thank Hannah Kilpatrick and Mark George for their technical assistance.

## Author contributions

**Conceptualization:** Carly A. Lockard, Crystal Coolbaugh, Bruce M. Damon.

**Data curation:** Bruce M. Damon.

**Formal analysis:** Melissa T. Hooijmans, Xingyu Zhou.

**Funding acquisition:** Bruce M. Damon.

**Investigation:** Melissa T. Hooijmans, Crystal Coolbaugh.

**Methodology:** Melissa T. Hooijmans, Crystal Coolbaugh, Bruce M. Damon.

**Project administration:** Melissa T. Hooijmans.

**Resources:** Bruce M. Damon.

**Software:** Xingyu Zhou, Bruce M. Damon.

**Supervision:** Bruce M. Damon.

**Writing – original draft:** Melissa T. Hooijmans, Bruce M. Damon.

**Writing – review & editing:** Melissa T. Hooijmans, Carly A. Lockard, Xingyu Zhou, Crystal Coolbaugh, Roberto P. Guzman, Mariana E. Kersh, Bruce M. Damon.

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
