## [Decision Letter · Decision Letter 0]

30 Aug 2024

Dear Dr. Hooijmans,

Thank you for submitting your manuscript to PLOS ONE. After careful consideration, we feel that it has merit but does not fully meet PLOS ONE’s publication criteria as it currently stands. Therefore, we invite you to submit a revised version of the manuscript that addresses the points raised during the review process.

We look forward to receiving your revised manuscript.

Kind regards,

Charlie M. Waugh

Academic Editor

PLOS ONE

Journal requirements: 1. When submitting your revision, we need you to address these additional requirements. Please ensure that your manuscript meets PLOS ONE's style requirements, including those for file naming. The PLOS ONE style templates can be found at https://journals.plos.org/plosone/s/file?id=wjVg/PLOSOne_formatting_sample_main_body.pdf and https://journals.plos.org/plosone/s/file?id=ba62/PLOSOne_formatting_sample_title_authors_affiliations.pdf. 2. Thank you for stating the following in the Acknowledgments Section of your manuscript: [The authors thank Hannah Kilpatrick and Mark George for their technical assistance.  BD acknowledges NIH grants NIH R01 AR073831 and NIH S10 OD021771. The sponsor did not play a role in the study design, data collection and analysis, decision to publish or preparation of the manuscript.]We note that you have provided funding information that is not currently declared in your Funding Statement. However, funding information should not appear in the Acknowledgments section or other areas of your manuscript. We will only publish funding information present in the Funding Statement section of the online submission form. Please remove any funding-related text from the manuscript and let us know how you would like to update your Funding Statement. Currently, your Funding Statement reads as follows:  [BD acknowledges NIH grants NIH R01 AR073831 and NIH S10 OD021771; https://grants.nih.gov/. The sponsor did not play a role in the study design, data collection and analysis, decision to publish or preparation of the manuscript. ] Please include your amended statements within your cover letter; we will change the online submission form on your behalf. 3. Please expand the acronym “NIH” (as indicated in your financial disclosure) so that it states the name of your funders in full.This information should be included in your cover letter; we will change the online submission form on your behalf. 4. We note that you have indicated that there are restrictions to data sharing for this study. For studies involving human research participant data or other sensitive data, we encourage authors to share de-identified or anonymized data. However, when data cannot be publicly shared for ethical reasons, we allow authors to make their data sets available upon request. For information on unacceptable data access restrictions, please see http://journals.plos.org/plosone/s/data-availability#loc-unacceptable-data-access-restrictions.  Before we proceed with your manuscript, please address the following prompts: a) If there are ethical or legal restrictions on sharing a de-identified data set, please explain them in detail (e.g., data contain potentially identifying or sensitive patient information, data are owned by a third-party organization, etc.) and who has imposed them (e.g., a Research Ethics Committee or Institutional Review Board, etc.). Please also provide contact information for a data access committee, ethics committee, or other institutional body to which data requests may be sent. b) If there are no restrictions, please upload the minimal anonymized data set necessary to replicate your study findings to a stable, public repository and provide us with the relevant URLs, DOIs, or accession numbers. Please see http://www.bmj.com/content/340/bmj.c181.long for guidelines on how to de-identify and prepare clinical data for publication. For a list of recommended repositories, please see https://journals.plos.org/plosone/s/recommended-repositories. You also have the option of uploading the data as Supporting Information files, but we would recommend depositing data directly to a data repository if possible. Please update your Data Availability statement in the submission form accordingly. 5. Please include your full ethics statement in the ‘Methods’ section of your manuscript file. In your statement, please include the full name of the IRB or ethics committee who approved or waived your study, as well as whether or not you obtained informed written or verbal consent. If consent was waived for your study, please include this information in your statement as well. 

Reviewers' comments:

Reviewer's Responses to Questions

**Comments to the Author**

1. Is the manuscript technically sound, and do the data support the conclusions?

Reviewer #1: Yes

Reviewer #2: Yes

2. Has the statistical analysis been performed appropriately and rigorously?

Reviewer #1: Yes

Reviewer #2: Yes

3. Have the authors made all data underlying the findings in their manuscript fully available?

Reviewer #1: Yes

Reviewer #2: No

4. Is the manuscript presented in an intelligible fashion and written in standard English?

Reviewer #1: Yes

Reviewer #2: Yes

Reviewer #1: This manuscript explores a new method for tracking changes in muscle architecture using DTI as muscles shorten passively. The study proposes a registration strategy that helps translate DTI data between different muscle states ( from relaxed to a contracted) by focusing on muscles during joint movements (plantarflexion and dorsiflexion). The researchers tested their approach on seven healthy individuals, using advanced imaging and computational techniques to see how well the method could predict muscle changes. The study is of high scientific value since there is no sufficient knowledge regarding muscle DTI parameters in voluntarily contracted muscles.

The registration strategy introduced here could be a promising way to model muscle behavior during contractions without the need of direct imaging, which could be particularly useful in biomechanic analysis of muscle behaviour with MRI.

The manuscript is well written and the study design is sound. There are no minor or major issues to consider.

Reviewer #2: Introduction:

- Line 91: It would be helpful to add a brief sentence at the end of the last Introduction paragraph that summarizes the overall aim and objective of this study.

Materials and Methods:

- Lines 107-110: The described setup would be clearer if you provided a corresponding schematic figure that shows a visualization of the positioning in the scanner.

- Line 112: Why were the two angles (+20° and -10°) chosen? I understand that these angles are within the device limits but is there functional relevance for these two angles? Why were the chosen angles not symmetrically displaced from neutral position (i.e., +10° and -10°)? It would be nice to explain briefly either here or in the introduction.

- Figure 1: Please label the TA masks (the TA muscle and the TA aponeurosis). There should also be an arrow from both the DTI image and the TA anatomical masks that lead into the fiber tracts image, because the mask is used to define the tractography parameters. The current figure makes it seem like the fiber tracking is completely independent from the masks. Additionally, it would be helpful to label (1), (2), and (3) on the bottom of the three fiber-tracts images, to correspond to the three sets of fiber-tracts described in lines 182-189. Labels would be helpful here as well (i.e., (1) Original +20°, (2) Original -10°), and (3) Transformed -10°). Finally, in general, this figure is a bit confusing and it would be helpful if there were more text and labels added so the analysis pipeline is easily understandable without having to constantly refer to the text.

- Lines 153-156: How did you come up with this selection of registration inputs to explore? Is this based on previous literature, and if so, can you provide references?

- Line 165-166: Can you explain further what the Sorenson Dice Similarity Coefficient (DSC) is by providing an equation, description, or reference please?

- Lines 198-199: How did you come up with the exclusion criteria used in the fiber_goodness() function? Can you please provide an explanation or a reference?

- Lines 206-207: How did you define the deep versus superficial compartment?

Results:

- Figure 3: It is a bit hard to see the red mask overlay on the MR image on the left side — can you please make these a bit darker or less transparent?

- Figure 4: The results section says there is a significant difference in pennation angles between joint positions, but this is not labelled on the figure. Please label with an asterisk and p-value.

- Figures 3-4: It would be helpful to label "ns" and p-values on each panel so don't need to refer back to the text to know which ones were non-significant.

- Supplementary Figure 1: How are “high” and “low” similarity values defined? Maybe you can state what the similarity values are for each in the figure legend, as in Figure 6.

- Figures (general): I noticed you use “fibre” in the figures but “fiber” in the manuscript body. Please choose one for consistency.

- Line 283: Please report the r and p values for delta curvature.

Discussion

- Lines 317-318: What position was the ankle at in these previous studies? How are the angles studied in this study similar?

- Lines 326-329: Expand on this explanation please. Is the smaller pennation angle due to the TA being shortened during plantarflexion?

- Line 329-335: You explain the lack of differences in fascicle length between ankle positions, but it would be nice to add an explanation for the lack of significant difference in curvature as well.

- Line 344: Can you give an estimate for the “higher internal contrast” in the OP images versus the other images (for example, the CNR measures in each case)? This could be added briefly here or in a more structured format in the results section.

- Lines 357-360: This sentence is a bit confusing. If the similarity measure takes both parameters into account, then wouldn't fiber tract pairs that lay further apart have a lower similarity value than adjacent fiber-tract pairs? Please clarify this.

- Line 372: If similarity measures cannot be used for the long-term goal (dynamic contractions), why was it used here as a validation metric? This paragraph reads as a bit contradictory to the objective of this study, as it sounds like the end goal of this work is not related to the validation and analysis presented in the current study. A further explanation or clarification in this paragraph would be helpful.

- How does noise affect the registration of images? I know denoising is included in the processing pipeline, but were the two images of similar SNR in each participant to ensure the most accurate and robust transformation? If not, this should be explained as a limitation.

**Do you want your identity to be public for this peer review?** For information about this choice, including consent withdrawal, please see our Privacy Policy

Reviewer #1: No

Reviewer #2: No

---

## [Author Response · Author response to Decision Letter 1]

24 Oct 2024

Response to the reviewers

The authors thank the referees for carefully reviewing our manuscript and for the valuable suggestions to improve our manuscript. We addressed the comments in the revised manuscript and answered each of the comments below. The portions of the text that were modified in response to the reviewers’ comments are indicated with comment bubbles. We have also edited the manuscript for clarity, usage, and brevity.

Reviewer #1: This manuscript explores a new method for tracking changes in muscle architecture using DTI as muscles shorten passively. The study proposes a registration strategy that helps translate DTI data between different muscle states ( from relaxed to a contracted) by focusing on muscles during joint movements (plantarflexion and dorsiflexion). The researchers tested their approach on seven healthy individuals, using advanced imaging and computational techniques to see how well the method could predict muscle changes. The study is of high scientific value since there is no sufficient knowledge regarding muscle DTI parameters in voluntarily contracted muscles.

The registration strategy introduced here could be a promising way to model muscle behavior during contractions without the need of direct imaging, which could be particularly useful in biomechanic analysis of muscle behaviour with MRI. The manuscript is well written and the study design is sound. There are no minor or major issues to consider.

Thank you for taking the time to review our manuscript and for your kind words.

Reviewer #2: Introduction:

- Line 91: It would be helpful to add a brief sentence at the end of the last Introduction paragraph that summarizes the overall aim and objective of this study.

R2.1

Thank you for the suggestion. We have included this at the end of the introduction.

Added text:

“ The overall aim of this study was to evaluate if an image registration strategy could be used to convert the whole-muscle average architectural properties from an extended joint position to those of a flexed position.”

Materials and Methods:

- Lines 107-110: The described setup would be clearer if you provided a corresponding schematic figure that shows a visualization of the positioning in the scanner.

R2.2

We have added a schematic image of the set-up in the scanner as supplemental material.

Supplemental Figure legend:

Schematic overview of the positioning in the MRI scanner for the two ankle positions. In green is shown the anterior receive coil positioned on top of the lower extremities.

- Line 112: Why were the two angles (+20° and -10°) chosen? I understand that these angles are within the device limits but is there functional relevance for these two angles? Why were the chosen angles not symmetrically displaced from neutral position (i.e., +10° and -10°)? It would be nice to explain briefly either here or in the introduction.

R2.3

We selected these values because they represented the maximum comfortable achievable positions for most participants, aligning with the typical range of motion expected during dynamic exercise. Symmetrical angles were not used, as the comfortable limit to the range of motion for plantarflexion tends exceed that of dorsiflexion.

- Figure 1: Please label the TA masks (the TA muscle and the TA aponeurosis). There should also be an arrow from both the DTI image and the TA anatomical masks that lead into the fiber tracts image, because the mask is used to define the tractography parameters. The current figure makes it seem like the fiber tracking is completely independent from the masks. Additionally, it would be helpful to label (1), (2), and (3) on the bottom of the three fiber-tracts images, to correspond to the three sets of fiber-tracts described in lines 182-189. Labels would be helpful here as well (i.e., (1) Original +20°, (2) Original -10°), and (3) Transformed -10°). Finally, in general, this figure is a bit confusing and it would be helpful if there were more text and labels added so the analysis pipeline is easily understandable without having to constantly refer to the text.

R2.4

We have adjusted the figure and incorporated the changes suggested by the reviewer, and hope that provides sufficient clarity of presentation.

Adjusted figure:

Adjusted figure legend:

“MR data acquired in two ankle positions, +20° (blue outline) and -10° (green outline). Anatomical images in the two ankle positions were manual segmented to derive masks for the TA aponeurosis (-10°) and TA (-10° and +20°) muscle. After the tensor calculation (+20°), fiber tracking was performed using the TA mask and aponeurosis mesh (+20°) as input. Resulting in the (1) fiber-tracts for the +20° position. Displacement fields derived from registration of the anatomical images (+20° and -10°) were used to transform fiber-tracts and the aponeurosis mesh from +20° to -10° ankle position (orange arrows), leading to the (2) transformed fiber-tracts. The transformed aponeurosis in combination with the manually segmented TA muscle mask (-10°) and tensor (-10°)(green arrows) are used to initiate tracking in the -10° position generating the (3) fiber-tracts for the +20° position. These inputs are required to allow direct comparison between (2) transformed (orange dotted outline) and (3) original fiber-tracts (green dotted outline).”

- Lines 153-156: How did you come up with this selection of registration inputs to explore? Is this based on previous literature, and if so, can you provide references?

R2.5

The selection of registration inputs is based on the author’s prior experience with similar datasets rather than on published literature. As a result, no specific references can be provided.

- Line 165-166: Can you explain further what the Sorenson Dice Similarity Coefficient (DSC) is by providing an equation, description, or reference please?

R2.6

We have added a description and reference Sorenson Dice Similarity Coefficient.

Adjusted text:

For each set of registration inputs, the outcomes were assessed by calculating the Sorenson Dice Similarity Coefficient (DSC), a similarity quotient that ranges between zero and one and is calculated as:

DSC= (A∩B)⁄((|A|+|B|) )

where A is set of points in the mask of image A, B is set B, ∩ indicates the intersection of the sets, and |┤| indicates the size of the set. The DSC was calculated for the muscle mask (DSCmask) and aponeurosis (DSCapo) mask. In addition, the registration outcomes were assessed using the Hausdorff distance (DH) and Euclidean distance (DE) for the aponeurosis mesh.

- Lines 198-199: How did you come up with the exclusion criteria used in the fiber_goodness() function? Can you please provide an explanation or a reference?

Response 2.7

The exclusion criteria used for the fiber goodness () function are derived from findings on fiber length, pennation angle and curvature reported in literature from ultrasound studies. We have rephrased the text and hope that this communicates the procedure and rationale more effectively.

Adjusted text:

“The maximum length criterion was set at the average fascicle length plus twice the standard deviation, reported in an ultrasonography study (54). For pennation angle and curvature, values were selected to ensure the exclusion of physiologically unrealistic or otherwise implausible data.”

- Lines 206-207: How did you define the deep versus superficial compartment?

Response 2.8

The distinction between the deep and superficial compartment of the TA is made by visually identifying the internal aponeurosis.

Results:

- Figure 3: It is a bit hard to see the red mask overlay on the MR image on the left side — can you please make these a bit darker or less transparent?

Response 2.9

Thank you for pointing this out. We have adjusted the level of transparency to make the mask overlay easier to detect. See adjusted figure response 2.11.

- Figure 4: The results section says there is a significant difference in pennation angles between joint positions, but this is not labelled on the figure. Please label with an asterisk and p-value.

Response 2.10

We have added the p-value and asterisk to the figure.

Adjusted Figure:

- Figures 3-4: It would be helpful to label "ns" and p-values on each panel so don't need to refer back to the text to know which ones were non-significant.

Response 2.11

We have added the ns and p-values to each of the panels.

Adjusted Figure:

- Supplementary Figure 1: How are “high” and “low” similarity values defined? Maybe you can state what the similarity values are for each in the figure legend, as in Figure 6.

Response 2.12

Thank you for the suggesting. We have slightly rephrased the figure legend to clarify our approach and have also included the average similarity value for each of datasets presented.

Adjusted figure legend:

Bland-Altman Plots showing the similarity between the original and transformed fiber-tracts for pennation angle (Δθ) and fiber-tract length (ΔLFT), on a fiber tract level, for the dataset with on average the highest similarity value (left) and for a dataset with on average the lowest similarity (right) value (Si = 0.14 & 0.31). Note differences in Y-axis scales between the left and right panels.

- Figures (general): I noticed you use “fibre” in the figures but “fiber” in the manuscript body. Please choose one for consistency.

Response 2.13

We apologize for the inconsistency and have revised the wording throughout the manuscript for clarity and consistency.

- Line 283: Please report the r and p values for delta curvature.

Response 2.14

We have added the r and p values for the delta curvature to the result section.

Discussion

- Lines 317-318: What position was the ankle at in these previous studies? How are the angles studied in this study similar?

Response 2.15

We have included information on the ankle angles used in the previous studies referenced in the discussion section of the manuscript. In those studies, the ankle angles were slightly larger, with -15 for dorsiflexion and +15 and +30 for plantarflexion.

Adjusted text:

”previous observations that showed a smaller pennation angle in plantarflexion (+15 and +30°) compared to dorsiflexed (-15°) position(54, 62, 63)”

- Lines 326-329: Expand on this explanation please. Is the smaller pennation angle due to the TA being shortened during plantarflexion?

Response 2.16

During plantarflexion, the TA muscle is lengthened, not shortened. We have expanded the explanation in the discussion section to clarify this point.

Adjusted text:

“Additionally, we observed a smaller pennation angle in the full TA in the plantarflexed (muscle lengthening) (+20°) compared to dorsiflexed (muscle shortening) (-10°) ankle position, which is consistent with physiological predictions based on the relative compliance of resting muscle vs. tendinous structures and previous observations that showed a smaller pennation angle in plantarflexion (+15 and +30°) compared to dorsiflexed (-15°) position(54, 62, 63).”

- Line 329-335: You explain the lack of differences in fascicle length between ankle positions, but it would be nice to add an explanation for the lack of significant difference in curvature as well.

Response 2.17

We have added a detailed explanation to the discussion section of the manuscript to address the lack of significant difference in curvature.

Added text:

“Even though physiological predictions suggested higher curvature, in the dorsiflexed position, we did not observe a clear difference in this property at the group level. The failure to observe a significant change could have occurred due to insufficient statistical power for the study or if the curvature estimation method was insufficiently precise. Additional work may be necessary to improve the accuracy and precision of this method. ”

- Line 344: Can you give an estimate for the “higher internal contrast” in the OP images versus the other images (for example, the CNR measures in each case)? This could be added briefly here or in a more structured format in the results section.

Response 2.18

We cannot provide a precise estimate, but the issue is likely related to the water image being derived from the original echo images (IP and OP). In the first OP image, a shorter echo time results in stronger signal form tissue types with short relaxation characteristics, such as collagen. Relying solely on this image may have enhanced the internal contrast-to-noise ratio (CNR).

- Lines 357-360: This sentence is a bit confusing. If the similarity measure takes both parameters into account, then wouldn't fiber tract pairs that lay further apart have a lower similarity value than adjacent fiber-tract pairs? Please clarify this.

Response 2.19

Thank you for pointing this out. Yes, that is how the similarity measure should work. We have adjusted the text in the discussion section manuscript to clarify.

Adjusted text: “This may be because the similarity measure accounts for both the amount of length overlap of two tracts and the distance between their corresponding points. Depending on the value chosen for the weighting coefficient, differently shaped but adjacent fiber-tracts and overlapping but more distant fiber-tracts can have the same similarity score.”

- Line 372: If similarity measures cannot be used for the long-term goal (dynamic contractions), why was it used here as a validation metric? This paragraph reads as a bit contradictory to the objective of this study, as it sounds like the end goal of this work is not related to the validation and analysis presented in the current study. A further explanation or clarification in this paragraph would be helpful.

Response 2.20

Thank you for highlighting this point. In this study, our goal was to validate the registration strategy as a tool for transforming fibers, making it crucial to compare it with ground-truth data, something that had not been done before. However, the intended end-application of this approach will be in scenarios where acquiring ground-truth data is impossible, and thus direct validation won’t be possible. We have revised the text slightly to clarify this distinction.

Adjusted text:

“The future goal of this approach is to explore muscle architecture during contractions.”

- How does noise affect the registration of images? I know denoising is included in the processing pipeline, but were the two images of similar SNR in each participant to ensure the most accurate and robust transformation? If not, this should be explained as a limitation.

Response 2.21

The two images used for registration were acquired under the same settings and in close succession, so we do not expect significant changes in SNR between them. Additionally, all data were visually inspected for artifacts, movement or notable changes in quality prior to analysis and no datasets were excluded. Based on this, we are confident that the two images are of similar quality.

---

## [Decision Letter · Decision Letter 1]

8 Nov 2024

A registration strategy to characterize DTI-observed changes in skeletal muscle architecture due to passive shortening

PONE-D-24-14095R1

Dear Dr. Hooijmans,

We’re pleased to inform you that your manuscript has been judged scientifically suitable for publication and will be formally accepted for publication once it meets all outstanding technical requirements.

Kind regards,

Charlie M. Waugh

Academic Editor

PLOS ONE

Additional Editor Comments (optional):

Reviewers' comments:

Reviewer's Responses to Questions

**Comments to the Author**

Reviewer #2: All comments have been addressed

2. Is the manuscript technically sound, and do the data support the conclusions?

Reviewer #2: Yes

3. Has the statistical analysis been performed appropriately and rigorously?

Reviewer #2: Yes

4. Have the authors made all data underlying the findings in their manuscript fully available?

Reviewer #2: Yes

5. Is the manuscript presented in an intelligible fashion and written in standard English?

Reviewer #2: Yes

Reviewer #2: (No Response)

**Do you want your identity to be public for this peer review?** For information about this choice, including consent withdrawal, please see our Privacy Policy

Reviewer #2: No

---

## [Editor Report · Acceptance letter]

PONE-D-24-14095R1

PLOS ONE

Dear Dr. Hooijmans,

I'm pleased to inform you that your manuscript has been deemed suitable for publication in PLOS ONE. Congratulations! Your manuscript is now being handed over to our production team.

Kind regards,

on behalf of

Dr. Charlie M. Waugh

Academic Editor

PLOS ONE